# Break the Visual Perception: Adversarial Attacks Targeting Encoded Visual Tokens of Large Vision-Language Models

## ABSTRACT

Large vision-language models (LVLMs) integrate visual information into large language models, showcasing remarkable multi-modal conversational capabilities. However, the visual modules introduces new challenges in terms of robustness for LVLMs, as attackers can craft adversarial images that are visually clean but may mislead the model to generate incorrect answers. In general, LVLMs rely on vision encoders to transform images into visual tokens, which are crucial for the language models to perceive image contents effectively. Therefore, we are curious about one question: Can LVLMs still generate correct responses when the encoded visual tokens are attacked and disrupting the visual information? To this end, we propose a non-targeted attack method referred to as **VT-Attack** (Visual Tokens Attack), which constructs adversarial examples from multiple perspectives, with the goal of comprehensively disrupting feature representations and inherent relationships as well as the semantic properties of visual tokens output by image encoders. Using only access to the image encoder in the proposed attack, the generated adversarial examples exhibit transferability across diverse LVLMs utilizing the same image encoder and generality across different tasks. Extensive experiments validate the superior attack performance of the VT-Attack over baseline methods, demonstrating its effectiveness in attacking LVLMs with image encoders, which in turn can provide guidance on the robustness of LVLMs, particularly in terms of the stability of the visual feature space.

## CCS CONCEPTS

• **Computing methodologies** → **Artificial intelligence**; • **Security and privacy** → *Social aspects of security and privacy*.

## KEYWORDS

Large Vision-Language Model, Adversarial Attack, Image Encoder, Visual Tokens Attack

## 1 INTRODUCTION

Large vision-language models (LVLMs) have garnered considerable attention owing to their remarkable visual perception and language interaction capabilities [2, 38]. Compared to large language models (LLMs), LVLMs exhibit superiority in image understanding by leveraging visual models, making them highly effective for diverse multimodal tasks, such as image captioning [20, 21], visual question

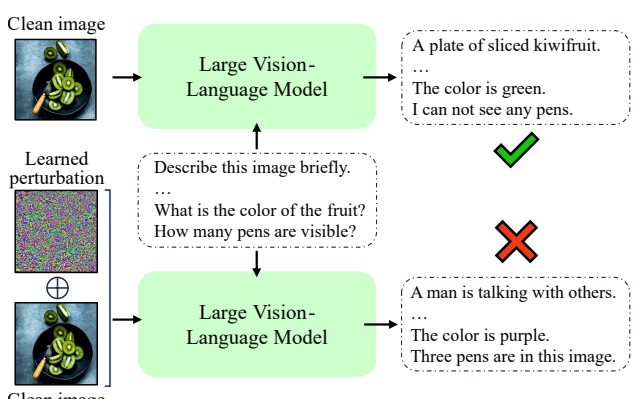

**Figure 1: An example of attacking LVLM. By introducing subtle perturbations to a clean image, the model fails to produce the correct answers. Despite using different prompts, the model is unable to generate correct outputs, indicating a breakdown in the effectiveness of visual information.**

answering [4] and multimodal dialogue [9, 23, 42]. However, recent research [5, 28, 41] highlights their vulnerability to adversarial attacks, posing potential security concerns in practical domains such as biological/medical image understanding [19] and document information extraction [22]. This underscores the importance of investigating the robustness of LVLMs from an attacker's perspective.

Adversarial attacks involve the deliberate manipulation of input data to induce incorrect or specific predictions. In general, adversarial attacks can be classified into targeted attacks, which aim to mislead the model into generating specific outputs, and non-targeted attacks, which lead to any incorrect or undesired outputs. Early research of adversarial attacks on visual models was primarily conducted to explore the security and robustness of image classification models [15, 25]. Attacking LVLMs is more challenging because of a much broader prediction space, where an image can correspond to multiple textual expressions while belonging to a specific class.

Extensive investigations have been conducted to explore the robustness of LVLMs, including targeted attacks [5, 28, 31, 32, 41] and non-targeted attacks [8, 31]. Specifically, the non-targeted attacks on LVLMs aim to mislead models into generating any erroneous answers, which raise security concerns as these attacks can potentially lead to the breakdown of models in practical scenarios. Therefore, it is vital to design effective non-targeted attacks to investigate the robustness of multimodal systems.

Existing non-targeted attacks commonly employ end-to-end [31] or CLIP-based [8] cross-modal optimization methods, aiming to deviate images from their original textual semantics. Nevertheless,

these approaches overlook the impact of manipulated visual tokens on the model robustness.

LVLMs typically employ ViTs [11] as image encoders to convert images into visual tokens, which encapsulate comprehensive image features/information, working as a bridge for subsequent modules (e.g. language modules) to perceive image contents. Essentially, disrupting visual tokens can impair the model's visual perception and ability to generate proper responses. Therefore, we suggest investigating the robustness of LVLMs by conducting adversarial attacks targeting the encoded visual tokens, providing assistance for relevant research in defense.

In this paper, we propose a multi-angle attack approach called **VT-Attack** (Visual Tokens Attack) which is designed to target the image encoder of LVLMs. As shown in Figure 2, our proposed approach consists of three sub-methods that systematically and comprehensively disrupt feature representations, inherent relationships, and global semantics of visual tokens output by the image encoder. This facilitates the exploration of the vulnerability of LVLMs to compromised visual information in the embedding space, simulating the operations of extreme adversaries in real-world scenarios.

Notably, our approach yields two benefits. Firstly, adversarial images crafted against the shared image encoder of LVLMs exhibit global effectiveness across different LVLMs [32]. Secondly, we find that the adversarial perturbations obtained through the image encoders of LVLMs are insensitive to specific prompts or tasks, as the generation process does not rely on the prompt/task information.

Methodologically, our method is applicable to LVLMs employing ViTs [11] as image encoder. We conduct experiments on a variety of prominent baseline LVLMs including LLaVA [23], MiniGPT-4 [42], LLaMA-Adapter-v2 [14], InstructBLIP [9], Otter [18], Open-Flamingo [3], BLIP-2 [21] and mPLUG-Owl-2 [37], with image encoders such as OpenAI CLIP [29], EVA CLIP [13] and other further-trained ViTs. An example of our attack is demonstrated in Figure 1. Empirical results demonstrate the effectiveness of our VT-Attack, consistently outperforming baseline approaches and individual sub-methods. Furthermore, employing adversarial examples generated against image encoders to attack downstream LVLMs can successfully mislead the models into generating incorrect answers, even with different questions as prompts. We also conduct experimental analyses on the properties and functions of each sub-method to demonstrate their distinct roles in attacking from different perspectives.

In summary, our contributions can be summarized as follows:

- We propose VT-Attack, a joint method that constructs adversarial images by disrupting the visual tokens output by image encoders of LVLMs from multiple perspectives, in order to investigate the robustness of LVLMs against compromised visual information.
- We conduct extensive experiments on various models to demonstrate the effectiveness of the proposed method. The results indicate that the adversarial images generated by our method exhibit cross-prompt generality and enhanced attack performance over baseline methods.
- We explore the distinctive properties and contributions of each sub-method in our attack approach through experimental analysis, validating the effectiveness of the joint method.

## 2 RELATED WORK

### 2.1 Large Vision-Language Models

Research in large vision-language models (LVLMs) has been advancing rapidly, driven by the efforts of researchers who design novel model architectures and employ specific training strategies to propel their development [3, 9, 14, 18, 20, 21, 23, 37, 42].

The architecture of a large vision-language model typically comprises three components: a pre-trained image encoder, an intermediate module facilitating the transformation of visual tokens into the language space, and a large language model. Various approaches have been employed in designing the intermediate modules. For instance, LLaVA [23] utilizes linear layers to project visual features into the language space, while the BLIP-2 [21] series (MiniGPT-4 [42], InstructBLIP [9]) adopt Q-Former to extract the most relevant visual features to the text prompts for the language models.

Different LVLMs may employ diverse intermediate modules for visual feature extraction, while utilizing a common pre-trained image encoder (e.g. OpenAI CLIP [29] or EVA CLIP [13]) for feature encoding. These pre-trained image encoders have been trained with contrastive learning on large-scale image-text datasets, allowing them to capture universal visual features that are beneficial for various downstream tasks.

### 2.2 Adversarial Attack

Adversarial attacks have been extensively explored to assess model robustness. Early research primarily focused on image classification, while in recent years, it has expanded into other domains [1, 33, 39, 40].

In general, adversarial attacks can be categorized into white-box attacks and black-box attacks, where white-box attacks allow attackers to have complete access to the model's architecture and parameters [12, 25]. In contrast, black-box attacks require attackers to launch attacks without any knowledge of the model's internal details [7, 27], making them intuitively more challenging. Because our proposed method only has access to image encoders, it can be classified into gray-box attacks, which involve partial access to the model's architecture and parameters.

Considerable research efforts have been devoted to developing novel algorithms for adversarial attacks [6, 15, 17, 25, 34], aiming to enhance the efficiency and imperceptibility of the attacks. These studies have contributed to the gradual improvement and refinement of adversarial attack methods.

### 2.3 Adversarial Robustness of LVLMs

With the expanding applications of adversarial attacks, researchers have initiated investigations into the adversarial robustness of LVLMs.

LVLMs are capable of performing various multimodal tasks, including image-text dialogue, detailed image description, and content explanation, presenting heightened challenges for adversarial attacks. Recent works have investigated the robustness of LVLMs. Among them, transferable adversarial examples are constructed

**Figure 2: Unified framework for VT-Attack. (a) Both the clean image and learnable adversarial image are fed into the image encoder, yielding the [CLS] token and encoded visual tokens. The objectives of the feature attack and relation attack are to perturb visual tokens away from their original feature representations while deviating from the original cluster centers they belong to. And the aim of the semantics attack is to increase the semantic discrepancy between an image and its caption texts. (b) We first utilize the image encoder to update the adversarial perturbation, inducing the disruption of the encoded visual features at multiple levels. Next, we feed the adversarial image into various LVLMs to execute the attack.**

using proxy models in [41], and methods such as gradient estimation are employed to attack LVLMs in black box settings. Malicious triggers are injected into the visual feature space to compromise the model security in [32]. The work in [8] conducts a comprehensive analysis on the robustness of LVLMs and devises a context-augmented image classification scheme to improve robustness. Other approaches utilize end-to-end gradient-based optimization methods to generate adversarial perturbations [5, 28, 31], typically with the cross-entropy loss as the objective to induce errors or achieve proximity between the output and a predefined target text.

Different from these works, we focus on investigating the robustness of LVLMs against impaired visual information encoded in visual tokens. We construct adversarial perturbations that disrupt visual features output by the image encoder from different perspectives, resulting in more comprehensive corruption of visual tokens and enhanced attack performance.

## 3 METHODOLOGY

In this section, we start with the problem formulation and then provide detailed explanations of our proposed method The framework of our method (VT-Attack) is shown in Figure 2, where we first construct adversarial examples against image encoders and proceed to attack LVLMs.

### 3.1 Problem Formulation

Let $F_\theta(x, q) \mapsto z$ denote a large visual language model parameterized by $\theta$, where $x$ is the input image and $q$ is the prompt input to the LVLM. Let $I_\phi$ denote the image encoder of the LVLM parameterized by $\phi$, which encodes images into visual tokens $v$. Additionally, let $M_\psi$ represent the intermediate module, parameterized by $\psi$, that processes visual tokens output by $I_\phi$ and transforms them into mapped visual tokens $p$:

$$v = I_\phi(x), \quad p = M_\psi(I_\phi(x))$$

Given the input prompt $q$ and the image $x$, the answer $z$ generated by LVLM can be represented as

$$z = F_\theta(M_\psi(I_\phi(x)), q)$$

Let $x_{\text{adv}} = x + \Delta_{\text{adv}}$ denote the adversarial image being constructed, exhibiting subtle differences $\Delta_{\text{adv}}$ from the clean image $x$. Our focus is on non-targeted attacks, where the adversarial image $x_{\text{adv}}$ leads the LVLM to generate any incorrect or unreasonable answers $\hat{y}$ different from the original answer $z$ as follows.

$$z \neq \hat{z} = F_\theta(M_\psi(I_\phi(x + \Delta_{\text{adv}})), q)$$

With only access to the parameters and gradients of the image encoder $I_\phi$, our method constructs adversarial images by setting optimization objectives based on the visual tokens $v$. During the generation process of $x_{\text{adv}}$, it is common to apply an $L_p$ norm constraint on the perturbation size, written as $\|x - x_{\text{adv}}\|_p = \|\Delta_{\text{adv}}\|_p \leq \epsilon$. It should be noted that setting $\epsilon$ to a large value may compromise the stealthiness of the generated adversarial images.

Original Image  Clustering Result  Original Image  Clustering Result

**Figure 3: The comparison of original images and clustering results, where tokens/patches belonging to the same cluster are displayed in the same color.**

## 3.2 Visual Feature Representation Attack

Large vision-language models commonly employ CLIP [29] as their image encoders which are typically based on the ViT architecture [11]. An input image is split into fixed-length patches, with each patch treated as a token and fed into the ViT.

Subsequently, the ViT encodes the image and generates a series of visual tokens $v$ arranged in an $L \times D$ matrix, which can be regarded as the visual feature representation of the image. After further integration of these visual tokens by the intermediate module, the language model can naturally generate outputs leveraging the visual information.

Hence, the features output by the image encoder provides crucial visual information to the entire model. Intuitively, if the visual features are disrupted and deviated from the original representation, subsequent modules will be unable to accurately interpret the image contents, leading to erroneous model outputs.

Motivated by this, we apply a visual feature representation attack as illustrated in Figure 2 (a), aiming to maximize the loss between the feature representation in visual tokens of the adversarial image and the original representation:

$$\max \quad \mathbb{E}\left[\sum_i \mathcal{L}(I_\phi(x_{\text{adv}})^i, I_\phi(x)^i)\right] \quad (1)$$

$$\text{s.t. } \|x - x_{\text{adv}}\|_p \le \epsilon$$

where $\mathcal{L}$ measures the difference or distance, which can be calculated using KL divergence or MSE. We employ the PGD [25] optimization algorithm to update the adversarial perturbations.

## 3.3 Visual Token Relation Attack

While the visual feature attack explicitly disrupts individual visual tokens, it may not fully consider the interdependencies among these tokens. Therefore, we introduce visual token relation attack.

The self-attention [36] layers in the ViT image encoder are responsible for capturing the relationships and dependencies among image patches or tokens, corresponding to the relevance between different regions in the image [11]. These layers enable the model to weigh the importance of each token in relation to the others, allowing for a comprehensive understanding of the image's contexts.

Therefore, within the visual tokens generated by the image encoder, each token tends to carry information of other tokens that have a higher degree of relationship with it. This enables the visual tokens to exhibit clustering properties, where tokens with higher correlation tend to be grouped together in the same cluster. As

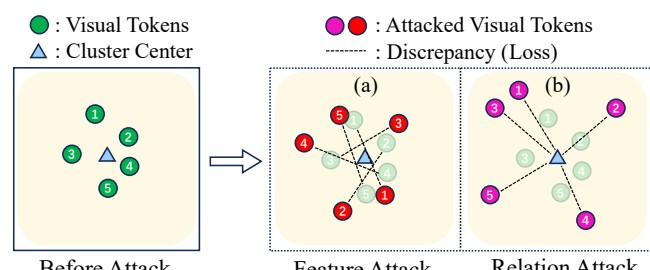

Before Attack  Feature Attack  Relation Attack

**Figure 4: An illustration of feature and relation attack. (a) and (b) demonstrate potential results of attacks based on feature and attacks based on relation, respectively.**

shown in Figure 3, visual tokens belonging to the same region or entity tend to cluster together. This clustering effect arises because tokens that are related or depict similar aspects of the image receive stronger attention connections through the self-attention mechanism.

Nevertheless, relying solely on the feature representation attack may not be sufficient to disrupt the clustering relationship among relevant visual tokens, as illustrated in Figure 4 (a). Although feature attack introduce deviations between visual tokens and their initial distribution, they may still exhibit proximity to the clustering centers.

To effectively disrupt the clustering relationships, we introduce a novel visual token relation attack, as illustrated in Figure 2 (a) and Figure 4 (b). Specifically, we initially apply the K-Means clustering [24] to the visual tokens generated by the image encoder, where the number of clusters $k$ is determined based on the silhouette coefficient [30]. Each visual token $v^i$ is assigned a cluster label denoted as $Y^i \in \mathcal{Y}$, and $k$ cluster centers are identified and denoted as $C$:

$$\mathcal{Y} = \{Y^1, \cdots, Y^L\}, C = \{C^1, \cdots, C^k\} \Leftarrow \text{Kmeans}(I_\phi(x)) \quad (2)$$

Next, we maximize the discrepancy between the visual tokens of the adversarial image and their respective cluster centers of the clean image:

$$\max \quad \mathbb{E}\left[\sum_i \mathcal{L}(I_\phi(x_{\text{adv}})^i, C^{Y^i})\right] \quad (3)$$

$$\text{s.t. } \|x - x_{\text{adv}}\|_p \le \epsilon$$

where $\mathcal{L}$ can be measured using KL divergence or MSE. By enlarging the discrepancy between visual tokens and the original cluster centers, adversarial images can disrupt the relationships among similar visual tokens. The disruption results in reduced dependencies among tokens within each cluster, causing visual tokens to contain less local adjacent feature information. Consequently, the subsequent modules of LVLMs struggle to effectively exploit the shared information among relevant tokens to comprehend the features of neighboring image patches.

**Algorithm 1** VT-Attack

**Require:**
    Image encoder $I_\phi$ of LVLM parameterized by $\phi$, input image $x$, image caption $t$, CLIP text encoder $T_\eta$, perturbation size $\epsilon$, updating rate $\alpha$, optimization steps $K$.

**Ensure:**
    Adversarial images $x_{\text{adv}}$;
1: Initialize $x_{\text{adv}} = x + \text{Clip}(\Delta_{\text{GaussianNoise}}, -\epsilon, \epsilon)$;
2: $\mathcal{Y} = \{Y^1, \cdots, Y^L\}, C = \{C^1, \cdots, C^k\} \Leftarrow \text{Kmeans}(I_\phi(x))$;
3: **for** $i = 1$ to $K$ **do**
4:     $\mathcal{L}_{\text{Feature}} = \mathbb{E}[\sum_i \mathcal{L}(I_\phi(x_{\text{adv}})^i, I_\phi(x)^i)]$
5:     $\mathcal{L}_{\text{Relation}} = \mathbb{E}[\sum_i \mathcal{L}(I_\phi(x_{\text{adv}})^i, C^{Y^i})]$
6:     $\mathcal{L}_{\text{Semantics}} = \mathcal{L}(I_\phi(x_{\text{adv}})^{[\text{CLS}]}, T_\eta(t)^{[\text{CLS}]})$
7:     $\mathcal{L} = \mathcal{L}_{\text{Feature}} + \mathcal{L}_{\text{Relation}} + \mathcal{L}_{\text{Semantics}}$
8:     Gradient descent: $x_{\text{adv}} = x_{\text{adv}} + \alpha \cdot \text{sign}(\nabla_{x_{\text{adv}}}(\mathcal{L}))$
9:     Perturbation size constraint: $x_{\text{adv}} = \text{Clip}_\epsilon(x_{\text{adv}})$
10:    Grayscale constraint: $x_{\text{adv}} = \text{Clip}(x_{\text{adv}}, 0, 1)$
11: **end for**
12: **return** $x_{\text{adv}}$;

## 3.4 Global Semantics Attack

The feature and relation attacks introduced above directly compromise the visual token sequence of length $L$ generated by the image encoder, resulting in disruptions at both the representation and relationship levels. While the combination of these two attacks can effectively disrupt the information of visual tokens, we are still interested in attacks that change the semantics of images.

The information carried by the [CLS] token contains the most direct content of an image, unlike the visual tokens that encode specific visual features. We hypothesize that the disruptions of both local image details (feature and relation attacks) and global semantics are mutually reinforcing, contributing to a comprehensive destruction of visual tokens.

Therefore, we incorporate the semantics attack as illustrated in Figure 2, which reduces the semantic similarity between the visual and text semantic information of the [CLS] token encoded by the CLIP image/text encoder:

$$\max \quad \mathcal{L}(I_\phi(x_{\text{adv}})^{[\text{CLS}]}, T_\eta(t)^{[\text{CLS}]}) \qquad (4)$$
$$\text{s.t. } \|x - x_{\text{adv}}\|_p \leq \epsilon$$

where $T_\eta$ represents the CLIP text encoder corresponding to $I_\phi$ and $t$ refers to the caption of an image. We utilize cosine similarity to preserve settings similar to contrastive learning [29]. We employ a variant of cosine similarity as the loss function $\mathcal{L}(\cdot, \cdot) = \frac{1}{1 + \text{cos\_sim}(\cdot, \cdot)}$ to align with the loss space of the previous two methods.

## 3.5 Visual Tokens Attack (VT-Attack)

By integrating the aforementioned three sub-attack methods, we introduce a unified attack approach named VT-Attack , as illustrated in Figure 2 (a). The proposed VT-Attack can comprehensively disrupt the embedded visual features, disturb the inherent relationships and weaken the semantic properties of visual tokens, by solving the following optimization problem:

$$\max \quad \mathcal{L}_{\text{Feature}} + \mathcal{L}_{\text{Relation}} + \mathcal{L}_{\text{Semantics}} \qquad (5)$$
$$\text{s.t. } \|x - x_{\text{adv}}\|_p \leq \epsilon$$

The generation process of the adversarial image is illustrated in Algorithm 1. After obtaining the adversarial image $x_{\text{adv}}$ through $I_\phi$, we input it to various LVLMs that utilize $I_\phi$ as the image encoder, as depicted in Figure 2 (b). Due to the models' inability to perceive meaningful visual information, they tend to generate incorrect answers regardless of the types of questions concerning the image content.

# 4 EXPERIMENTS

In this section, we present the experimental results of VT-Attack to demonstrate the effectiveness of the proposed method. Additionally, we provide experimental analysis of our approach for further exploration.

## 4.1 Experimental Settings

**Victim models.** We conduct experiments on a series of prominent baseline large vision-language models to validate the generality of our proposed method. The victim models include LLaVA [23], Otter [18], LLaMA-Adapter-v2 [14], and OpenFlamingo [3], which utilize OpenAI CLIP [29] as their image encoder, as well as BLIP-2 [21], MiniGPT-4 [42], and InstructBLIP [9], which employ EVA CLIP [13] as their image encoder. We also involve models without employing the pre-trained CLIP such as mPLUG-Owl-2 [37], utilizing a further trained ViT. We generate adversarial images using the ViT encoders and subsequently attack LVLMs that utilize the same image encoder.

**Dataset.** We follow the typical dataset construction in [10, 26, 31] by randomly sampling 1000 images from the validation set of ILSVRC 2012 for conducting adversarial attacks and evaluating robustness.

**Evaluation metric.** Following commonly used settings in [41], we employ the CLIP score for evaluating attack performance, which measures the similarity or alignment between images and texts. Both clean images $x$ and adversarial images $x_{\text{adv}}$ are fed into LVLMs to obtain clean and adversarial captions. Subsequently, we compute the CLIP scores between each image $x$ and the clean caption $z$ / the adversarial caption $\hat{z}$. The decrease in the CLIP score for the adversarial caption reflects the effectiveness of the attack. We also employ the attack success rate (ASR) used in [8] which represents the ratio of attacks that successfully mislead the model's output.

**Basic setup.** We follow the common setups in [8, 41], setting the maximum perturbation size $\epsilon$ to 8/255 and employing the infinity norm as the constraint [41]. Note that the images are normalized. For the feature attack and relation attack, we utilize the KL divergence or MSE to compute the losses $\mathcal{L}_{\text{Feature}}$ and $\mathcal{L}_{\text{Relation}}$. The PGD algorithm [25] with 1000 iterations is employed for optimization. For the relation attack, we determine the optimal number of clusters $k$ within a predefined interval using the silhouette coefficient [30].

## 4.2 Main Results

We conduct our evaluation primarily on the image captioning task [31, 41], as it assesses the global comprehension ability of

**Table 1: The results of VT-Attack on LVLMs. The evaluation metric is CLIP score(↓) which measures the similarity between images and clean captions or adversarial captions generated by LVLMs. The lower the score, the higher the degree of errors in the model's outputs, reflecting a better attack performance. "-" indicates that the attack cannot be executed due to the absence of a pre-trained text encoder. The gray background represents the attack results of the sub-methods. The best results are highlighted in bold. The best performance among the three sub-methods is highlighted in blue.**

| Image Encoder | OpenAI CLIP ViT | | | | EVA CLIP ViT | | | ViT (Trained) |
|---|---|---|---|---|---|---|---|---|
| Model | LLaVA | Otter-I | LLaMA Adapter-v2 | Open Flamingo | BLIP-2 | MiniGPT-4 | InstructBLIP | mPLUG Owl-2 |
| Clean | 31.94 | 30.87 | 31.49 | 31.95 | 30.44 | 32.45 | 31.07 | 32.61 |
| Random Tiny Noise | 31.65 | 30.84 | 31.33 | 31.83 | 30.29 | 32.26 | 31.13 | 32.55 |
| E2E [31] | 24.87 | 26.52 | 22.14 | 24.38 | 24.33 | 26.53 | 24.61 | 21.12 |
| CLIP-Based [8] | 23.24 | 20.04 | 20.19 | 18.53 | 21.56 | 21.72 | 21.47 | - |
| Semantics | 22.81 | 19.26 | 19.92 | 19.10 | 21.14 | 21.50 | 20.52 | - |
| Feature | 20.83 | 18.58 | 19.54 | 17.98 | 21.01 | 21.11 | 20.45 | 17.30 |
| Relation | 20.58 | 17.78 | 19.32 | 17.84 | 20.76 | 21.09 | 20.82 | 17.58 |
| VT-Attack (F+R) | 20.55 | 17.51 | 18.63 | **17.41** | 20.41 | 21.10 | **20.31** | **17.11** |
| VT-Attack | **20.33** | **16.76** | **18.18** | 17.48 | **20.32** | **20.64** | 20.47 | **17.11** |

LVLMs towards images. We query the LVLMs using the prompt `"Describe the image briefly in one sentence."` with adversarial images. The results are presented in Table 1. Note that "VT-Attack" refers to the combination of feature, relation and semantics attacks, while "VT-Attack (F+R)" refers to the combination of feature and relation attacks.

**Table 2: The attack successful rate across different tasks (question types) by VT-Attack. Here we evaluate Otter [18] as an example. Each task is evaluated using 10 prompts (Details are provided in supplementary materials). The ASR(↑) refers to the ratio of successful attacks that mislead the model's output. The best results are highlighted in bold.**

| Task | Image Caption | General VQA | Detailed VQA | Avg. |
|---|---|---|---|---|
| Tiny Noise | 0.089 | 0.114 | 0.133 | 0.112 |
| E2E [31] | 0.812 | 0.137 | 0.174 | 0.374 |
| CLIP-Based [8] | 0.851 | 0.289 | 0.635 | 0.592 |
| Semantics | 0.860 | 0.315 | 0.657 | 0.611 |
| Feature | 0.884 | 0.698 | 0.784 | 0.789 |
| Relation | 0.892 | 0.711 | 0.772 | 0.792 |
| VT-Attack (F+R) | 0.898 | **0.723** | 0.806 | 0.809 |
| VT-Attack | **0.914** | 0.715 | **0.828** | **0.816** |

As demonstrated in the results presented in Table 1, our proposed VT-Attack achieves the best attack performance. This indicates that our method can more extensively disrupt visual features compared to baseline methods, leading to a diminished comprehension of images by language models. Among the three sub-methods, the feature attack or relation attack typically achieves the best performance among the sub-methods, particularly the relation attack.

LVLMs exhibit stronger robustness against semantics attack compared to the other two methods. However, our analysis in Section 4.3 demonstrates the insensitivity of semantics attack to image complexity.

One can also notice that, adversarial examples generated by our proposed method exhibits transferability across the LVLMs employing same image encoder, as shown in Table 1. Regardless of the intermediate modules, the LVLMs exhibit vulnerability to visual tokens that lack original image information. Various intermediate modules fail to reconstruct the compromised visual content.

**Table 3: Cases of attack results against LLaVA in different methods.**

| Image | Method | LVLM-Output |
|---|---|---|
| | No Attack | a dog laying on the ground |
| | E2E [31] | a small puppy sitting on a fence |
| | CLIP-Based [8] | a cat and a dog playing together |
| | VT-Attack | a person holding a cellphone |
| | No Attack | a dessert on a plate |
| | E2E [31] | a pastry with chocolate sauce |
| | CLIP-Based [8] | a plate of food with ingredients |
| | VT-Attack | two people are standing together |

To further validate the generality of our method across various tasks or prompts, we conduct experiments using three different tasks, each evaluated with 10 prompts. Examples of the general VQA and the detailed VQA can be `"Is there a pen in the image?"` and `"Please provide a detailed description of the image"`. More prompts are provided in supplementary materials. We employ ASR for manual evaluation and the results are shown in Table 2. Compared to baseline methods and sub-methods,

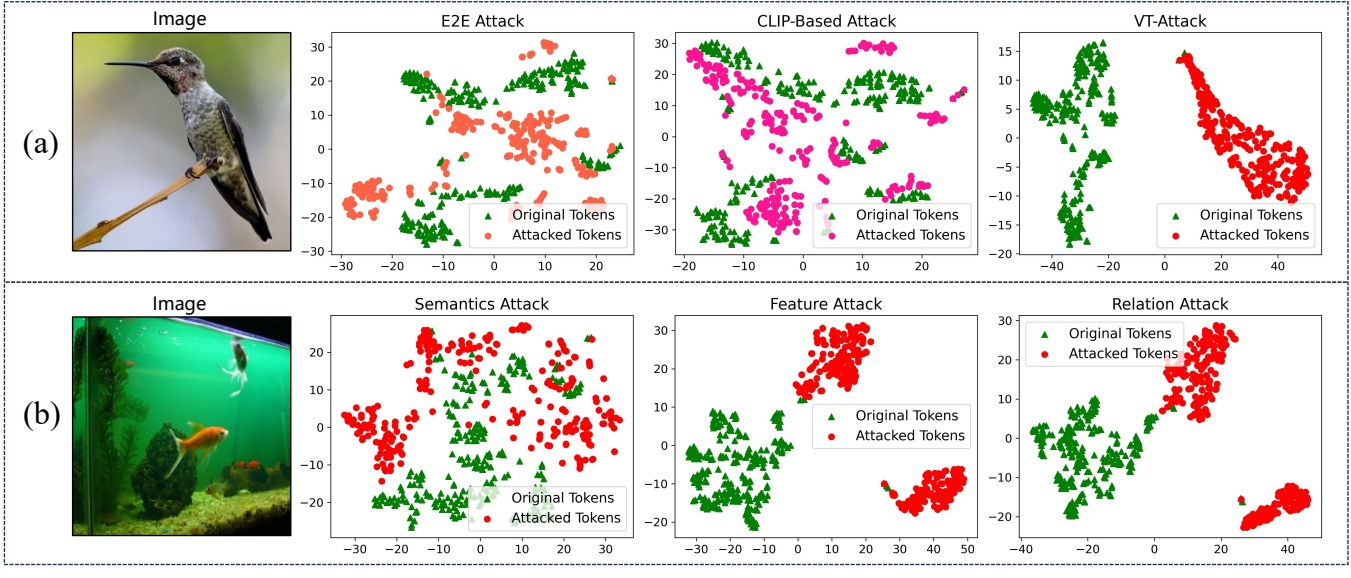

**Figure 5: Original image and the reduced-dimensional distribution of attacked visual tokens. (a) Comparison of attacked visual tokens between baseline methods and VT-Attack. (b) Comparison of attacked visual tokens among the three sub-methods of VT-Attack.**

VT-Attack achieves the highest ASR. We can observe that attacks against image encoders exhibit cross-task generality in contrast to end-to-end method [31]. This demonstrates the advantages of prompt-agnostic non-targeted attacks.

Table 3 presents cases of attacks on LLaVA [23]. Compared to the baseline methods, our attacks result in larger discrepancies between the model's outputs and the golden captions. More cases are provided in supplementary materials. To compare the impact of VT-Attack with baseline methods on the encoded visual tokens, we employ two-dimensional t-SNE [35] for visualization, as shown in Figure 5 (a). t-SNE is a dimensionality reduction technique that visualizes data in a lower-dimensional space, while preserving the local structure between data points. In contrast to the baseline methods, the visual tokens perturbed by VT-Attack exhibit a significant deviation from the original distribution. Such visual tokens likely have lost their original visual information, causing the language models to generate incorrect responses based on the image content. This demonstrates that VT-Attack targeting visual tokens can more effectively break the visual perception of LVLMs.

## 4.3 Sub-Method Analysis

**Visualization of visual tokens in three different sub-methods.**
In order to explore the degree of visual token disruption in each sub-method, we employ t-SNE [35] for visualization, as illustrated in Figure 5 (b). The attacked visual tokens produced by the semantics attack exhibit a distribution that remains relatively close to the original visual tokens. Nevertheless, the attacked visual tokens in the feature attack or relation attack have essentially deviated completely from the original distribution. This observation further explains why the performance of the single semantics attack is slightly lower.

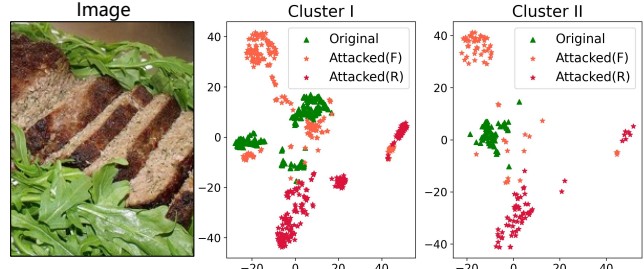

**Figure 6: Visualization of the original clusters and visual tokens in the feature attack (F) and relation attack (R).**

**Comparison of visual tokens in feature and relation attack.**
To identify the differences of attacked visual tokens between the feature attack and relation attack, we conduct case visualization of these two attacks in the same t-SNE space. An example is illustrated in Figure 6. We can observe that the visual tokens affected by the feature attack may still remain close to the original cluster coverage. This indicates that the incorporation of the relation attack may enhance the efficacy of disrupting the relationships between visual tokens and the original cluster, thereby compensating for the limitations of feature attack.

**Semantics attack exhibit insensitivity to image complexity.**
The image complexity refers to the richness exhibited by objects, colors or entities within an image. Despite the obvious impact of the feature attack and relation attack on visual tokens, their performance is influenced by the image complexity, as we have observed in our experiments. This is because the increasing image complexity amplifies the intricacy of the information in visual tokens, leading to a decreased performance. We conduct a statistical analysis of the

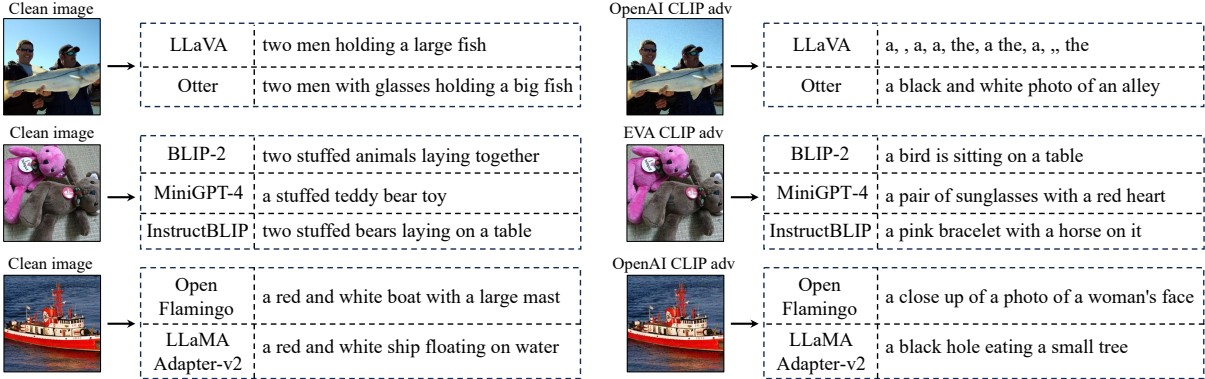

**Figure 7: Comparison of the responses generated by various LVLMs when queried with clean images and adversarial images.**

relationship between performance (ASR) and image complexity for the three sub-methods, as illustrated in Figure 8 (a). In contrast to the other two attack, the semantics attack is not sensitive to image complexity. A possible reason is that there is no direct correlation between image semantics and complexity because an image can be described concisely even if it exhibits complexity. Therefore, the semantics attack that disrupts semantic properties is insensitive to image complexity. The results demonstrates the advantage of employing semantics attack in mitigating the limitations of feature and relation attacks.

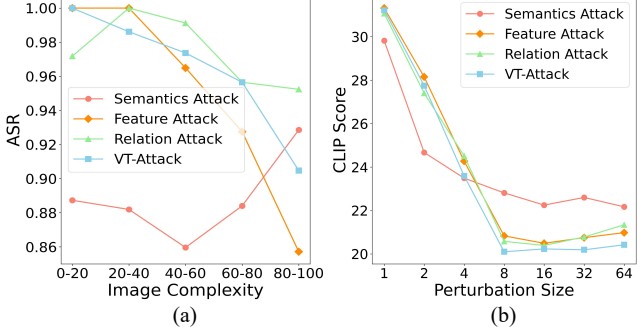

(a)                                (b)

**Figure 8: The influence of different conditions on attack performance against LLaVA. (a) ASR (↑) with respect to image complexity computed by segments of SAM [16]. (b) CLIP Score (↓) with respect to the perturbation size.**

### 4.4 Further Analysis

**The results of same adversarial image attacking different LVLMs.** We compare the answers generated by different LVLMs given the same clean and adversarial images, as shown in Figure 7. For the same clean image, the models produce similar answers. However, the models generate completely unrelated answers for the same adversarial image. The results indicate that adversarial images lack valid visual information that can be perceived by LVLMs. This can be attributed to the complete destruction of visual tokens,

**Table 4: Perplexity of output answers for different models queried with adversarial images.**

| Model | LLaVA | BLIP-2 | mPLUG-Owl-2 |
|---|---|---|---|
| Clean | 1.716 | 2.443 | 1.973 |
| Semantics | 2.165 | 3.261 | - |
| Feature | 4.287 | 2.999 | 2.362 |
| Relation | 4.580 | 3.026 | 2.376 |
| VT-Attack | 4.505 | 3.013 | 2.528 |

leading different models to rely on "guessing" in order to generate responses.

**The impact of perturbation size to the performance.** As shown in Figure 8 (b), the performance of the attack improves as the perturbation size increases from 1/255 to 8/255. However, further enlarging the perturbation size does not necessarily lead to performance improvement.

**The impact of attacks on the perplexity of model outputs.** We are curious about the self-confidence level of LVLMs in generating responses when the visual tokens are disrupted. Therefore, we compute the perplexity of the outputs from three models as shown in Table 4. The perplexity of answers corresponding to adversarial images is consistently higher than that of clean images. This indicates that the models exhibit vulnerability to compromised visual information, resulting in uncertain outputs.

## 5 CONCLUSION

In this paper, we focus on the robustness of LVLMs against non-targeted attacks. Existing methods often overlook the impact of compromised visual tokens on LVLMs. To this end, we propose a new adversarial attack method called VT-Attack, which disrupts the encoded visual tokens comprehensively from multiple perspectives. Experimental results have demonstrated the superiority of our approach over the baselines. This highlights the necessity for enhancing the adversarial defense capability of LVLMs. We hope our work can provide guidance for research on model defense, particularly in the defense of the visual token space.

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
