# OpenReview forum: "Break the Visual Perception: Adversarial Attacks Targeting Encoded Visual Tokens of Large Vision-Language Models"
_acmmm.org/ACMMM/2024/Conference — MM2024 Poster_

### Official Review · Reviewer_ZwA9 · 2024-05-16

**Rating:** 4
**Confidence:** 3

**Summary:**

This paper proposes VT-Attack, a joint method that constructs adversarial images by disrupting the visual tokens output by image encoders of LVLMs from multiple perspectives, i.e., feature, relation, and semantic. Extensive experiments are conducted to evaluate the attack performance of the proposed method and it can improve the performance compared to the existing methods.

**Strengths:**

1.The second sub-method visual token relation attack is interesting and innovative, which focuses on the inner relationship among visual tokens.

2.The experiments are solid and the analysis are reasonable.

**Limitations:**

1.What are the differences between visual tokens and feature embeddings?
The proposed visual tokens are the output of vision encoder, which seems similar to the image feature embeddings used in previous work [1]. What are the differences between visual tokens and feature embeddings? If they are the same, I think the proposed feature representation attack and global semantics attack are similar to the two attack strategies used in transfer-based attack in previous work [1].

2.From the experimental results, the visual token relation attack achieves the best attack performance. And it is a novel method to perform attack based on the clustering results of visual tokens. However, I have concerns about the optimization of equation 2 and 3. In my opinion, the process of equation 2 and 3 can change the clustering centers of visual tokens, but the visual tokens belonging to one cluster before the attack may lie in a new but same cluster, in this way, the relationship among similar visual tokens will not change.

3.According to the paper, the authors perform non-targeted attack and the clean images are selected from the validation set of ILSVRC 2012. However, it is unclear how the original caption 𝑡 used in equation 4 is obtained. This clarification is important for judging the success of the attack and for calculating the CLIP Score.

4.Does the [CLS] token in equation 4 come from the output of the vision encoder? The use of the [CLS] token from the text encoder is misleading because there is also a [CLS] token when tokenizing a text sequence. This makes it confusing how to calculate the semantic similarity between the visual and textual semantic information of the [CLS] token.

5.How is attack success defined? As presented in Table 3, can the outputs by E2E and CLIP-based methods be regarded as successful attack cases?

6.What encoders are used when calculating the CLIP Score? According to previous work [1], the average CLIP Score is about 40 even when using a clean image with an unrelated caption, while in Table 1 it is only about 30.

[1]. Zhao Y, Pang T, Du C, Yang X, Li C, Cheung NM, Lin M. On evaluating adversarial robustness of large vision-language models. Advances in Neural Information Processing Systems. 2024 Feb 13;36.

**Suitability:**

3

---

### Official Review · Reviewer_Wz5V · 2024-05-17

**Rating:** 4
**Confidence:** 4

**Summary:**

This paper propose a non-targeted attack method referred to as VT-Attack (Visual Tokens Attack), which constructs adversarial examples from multiple perspectives, with the goal of comprehensively disrupting feature representations and inherent relationships as well as the semantic properties of visual tokens output by image encoders.

**Strengths:**

1. The article proposes an attack method for encoding visual tokens, which is a novel perspective in evaluating the resilience of LVLMs.
2. The article describes four different attack methods, including visual feature representation attack, visual token relationship attack, global semantic attack, and visual token attack (VT Attack), demonstrating the threats to different levels of the model.
3. Through non target attacks, researchers emphasized that these attacks may lead to model failure in practical applications, thus emphasizing the importance of studying the robustness of LVLMs.

**Limitations:**

1. The article does not provide a detailed comparison with other known attack methods, which may limit readers' comprehensive understanding of the effectiveness of the new method.
2. Although the experimental setup was mentioned, the description of specific implementation details and parameter selection is not detailed enough, which may affect the reproducibility work of other researchers.
3. The article mainly focuses on attack methods, and there is relatively little discussion on how to enhance the resilience of LVLMs, namely defense strategies.

**Suitability:**

2

---

### Official Review · Reviewer_PUq6 · 2024-05-26

**Rating:** 5
**Confidence:** 3

**Summary:**

The paper introduces VT-Attack, a novel method designed to assess the robustness of LVLMs to compromised visual information. By systematically disrupting visual feature representations, interdependencies among visual tokens, and semantic attributes, VT-Attack comprehensively evaluates the vulnerability of LVLMs. Through extensive experiments, the proposed approach demonstrates its effectiveness in generating adversarial examples to mislead LVLMs.

**Strengths:**

1. The writing of the entire paper is very well done and easy to understand. The experiments are comprehensive.
2. The introduction of the relation attack is innovative, and the authors have conducted numerous ablation studies and visualizations to validate each type of attack proposed in the method.
3. The division of different aspects into "semantics, general features, and feature central relationships" proposed in this paper is reasonable.

**Limitations:**

questions:
1. It's interesting to see why Feature scored lower compared to relation in the experiment, it stands to reason that if you distance the token from the clustering center token it might be more effective than feature as a whole? In Figure 8, we can see that the average performance of relation attack is higher than the set of three sub-methods "VT-Attack", why?
2. For Table 1 using clip scores to calculate the success rate of attack I am more concerned about the different models of data length is not the same this will lead to different clip scores, the author can statistics on the average length of the response of the different methods?
3. It would be beneficial to set additional metrics to evaluate the reliability of the proposed method, considering that the method is a non-targeted attack. For instance, an attack could be deemed successful if the model fails to mention the objects in the image after the attack. Currently, metrics like CLIP score and perplexity might not be sufficiently accurate.
4. What would the visualization of the attacked image tokens look like according to Figure 3?

**Suitability:**

3

---

### Official Review · Reviewer_XkW7 · 2024-05-29

**Rating:** 2
**Confidence:** 2

**Summary:**

Considering that VLMs use vision encoders to transform images into visual tokens, this paper investigates the robustness of VLMs against compromised visual information. This paper proposed an adversarial attack method, VT-Attack, disrupting the encoded visual tokens comprehensively from multiple perspectives. Experimental results towards various victim models (LLaVA, Otter, LLaMA-Adapter-v2, OpenFlamingo, BLIP-2, MiniGPT-4, and InstructBLIP) can demonstrate the effectiveness of the proposed method compared with the baselines.

**Strengths:**

+ The motivation is clear and the proposed method is interesting.
+ The paper is well-written and easy to follow.

**Limitations:**

- It seems that the proposed method only focused on vision encoders of VLMs. If the paper aims to investigate the robustness of VLMs, why not consider both vision attacks and language attacks?
- It seems that the current method would highly rely on similar vision encoders, so the attack style is just like white-box attacks. In addition,  in the experiments, only CLIP and EVA-CLIP are used as source encoders...
- The missing details. For example, why the paper did not use the recent VLMs, the most recent model is mPLUG_Owl_2 (Nov, last year). We don't know the version (1.5?) and the scale (7B?) of Llava... It seems that the reproducibility of the paper cannot be guaranteed.
- Lack of explanation. In Figure 5, it seems that the attacked tokens are separated into two parts. I was wondering why this happened and the connection to the proposed method.

**Suitability:**

3

---

### Meta-Review · Area_Chair_K9tf · 2024-07-01

**Recommendation:** Accept (Poster)
**Confidence:** 5

**Metareview:**

This paper received one 'Weak Reject', one 'Weak Accept', and two 'Borderline Accept' reviews. The rebuttal has addressed most of the concerns raised by all reviewers, except for two questions from Reviewer XkW7. Overall, I have checked the rebuttal and the original submission, we think this submission is a good candidate for ACM MM conference. We encourage the authors to include the new content and experiments mentioned in the rebuttal in the final version of the paper. Additionally, issues raised by reviewers XkW7 and ZwA9 in their final rating justifications need to be addressed and clarified in the final version.